# Impact of Prior Antibiotic Use in Primary Care on *Escherichia coli* Resistance to Third Generation Cephalosporins: A Case-Control Study

**DOI:** 10.3390/antibiotics10040451

**Published:** 2021-04-16

**Authors:** Chiara Fulgenzio, Marco Massari, Giuseppe Traversa, Roberto Da Cas, Gianluigi Ferrante, Richard Aschbacher, Verena Moser, Elisabetta Pagani, Anna Rita Vestri, Orietta Massidda, Peter Konstantin Kurotschka

**Affiliations:** 1Department of Public Health and Infectious Diseases, Postgraduate School of Medical Statistics and Biometry, University of Rome “La Sapienza”, 00185 Rome, Italy; Fulgenzio.chiara@gmail.com (C.F.); annarita.vestri@uniroma1.it (A.R.V.); 2Pharmacoepidemiology Unit, National Centre for Drug Research and Evaluation, Istituto Superiore di Sanità, 00161 Rome, Italy; marco.massari@iss.it (M.M.); G.Traversa@aifa.gov.it (G.T.); roberto.dacas@iss.it (R.D.C.); 3Agenzia Italiana del Farmaco, 00187 Rome, Italy; 4Azienda Ospedaliera, Universitaria Città della Salute e della Scienza di Torino, 10126 Turin, Italy; gianluigi.ferrante@cpo.it; 5Health Service of Bolzano/Bozen Province, 39100 Bolzano/Bozen, Italy; richard.aschbacher@sabes.it (R.A.); verena.moser@provincia.bz.it (V.M.); elisabetta.pagani@asbz.it (E.P.); 6Department of Cellular, Computational and Integrative Biology, University of Trento, 38123 Povo, Italy; orietta.massidda@unitn.it; 7Department of Medical Sciences and Public Health, Faculty of Medicine and Surgery, University of Cagliari, 09124 Cagliari, Italy

**Keywords:** drug-resistance, bacterial, anti-bacterial agents, primary care, *Escherichia coli*, cephalosporins, third generation cephalosporins, beta-lactamases, information storage and retrieval

## Abstract

Research is lacking on the reversibility of antimicrobial resistance (AMR). Thus, we aimed to determine the influence of previous antibiotic use on the development and decay over time of third generation cephalosporin (3GC)-resistance of *E. coli*. Using the database of hospital laboratories of the Autonomous Province of Bolzano/Bozen (Italy), anonymously linked to the database of outpatient pharmaceutical prescriptions and the hospital discharge record database, this matched case-control study was conducted including as cases all those who have had a positive culture from any site for 3GC resistant *E. coli* (3GCREC) during a 2016 hospital stay. Data were analyzed by conditional logistic regression. 244 cases were matched to 1553 controls by the date of the first isolate. Male sex (OR 1.49, 95% CI 1.10–2.01), older age (OR 1.11, 95% CI 1.02–1.21), the number of different antibiotics taken in the previous five years (OR 1.20, 95% CI 1.08–1.33), at least one antibiotic prescription in the previous year (OR 1.92, 95% CI 1.36–2.71), and the diagnosis of diabetes (OR 1.57, 95% CI 1.08–2.30) were independent risk factors for 3GCREC colonization/infection. Patients who last received an antibiotic prescription two years or three to five years before hospitalization showed non-significant differences with controls (OR 0.97, 95% CI 0.68–1.38 and OR 0.85, 95% CI 0.59–1.24), compared to an OR of 1.92 (95% CI 1.36–2.71) in those receiving antibiotics in the year preceding hospitalization. The effect of previous antibiotic use on 3GC-resistance of *E. coli* is highest after greater cumulative exposure to any antibiotic as well as to 3GCs and in the first 12 months after antibiotics are taken and then decreases progressively.

## 1. Introduction

The introduction of penicillin in the 1930–1940s initiated the antibiotic era, which contributed significantly to the global decrease of morbidity and mortality due to communicable diseases [1]. Antimicrobial resistance (AMR), likely due to natural selection occurring when microorganisms are exposed to antimicrobial drugs, emerged shortly afterwards [2,3]. This prompted the search and development of new effective antibiotics but, since the antibiotic pipeline is now running dry, the world is facing the threat of a post-antibiotic era. In addition to AMR, the excessive use of antibiotics has been recently reported to have consequences beyond the clinical setting, the impact of which is only beginning to be appreciated [3]. Therefore, reducing unnecessary antibiotic use is a pivotal strategy to preserve the efficacy of these drugs and also to reduce unintended effects [4,5].

The health burden of AMR is of global concern, since it causes longer hospital stays and higher healthcare and societal costs. It is estimated that every year in Europe 33,000 people die due to infections caused by antibiotic resistant bacteria with approximately one third of these deaths (namely 10,762) occurring in Italy [6].

Although initially the occurrence of antibiotic resistant infections was deemed to be of concern especially in hospital settings, drug resistance of pathogens causing community acquired infections has emerged as a widespread phenomenon [7].

Consumption of antibiotics is extremely variable between and within countries: according to the European Surveillance of Antimicrobial Consumption Network (ESAC-net), community antibiotic use in 2018 ranged between 32.4 Defined Daily Doses (DDDs)/1000 inhabitants/day (DIDs) in Greece and 8.9 DIDs in the Netherlands [8]. A similar variability is observed within Italy, where, outpatient antibiotic use in 2018 varied between 23.0 DIDs in the Campania Region and 10.6 DIDs in the Autonomous Province of Bolzano/Bozen [9].

The highest burden attributable to infections caused by antibiotic resistant bacteria is determined by only a few species of resistant microorganisms. Among the list of the different species, third generation cephalosporin (3GC) resistant *Escherichia coli* (3GCREC) are top ranked, both in terms of number of cases and of attributable deaths, followed by methicillin-resistant *Staphylococcus aureus* and carbapenem-resistant *Pseudomonas aeruginosa* [6,10]. Time series analyses show how from 2002 and 2019 3GC *E. coli* resistance percentages increased significantly in Europe, with a population-weighted mean resistance of 15.1% in 2019 and large intercountry variability, reflecting a comparably high variation of antibiotic use in the hospital and in the community sector across countries [11,12]. This rising trend of 3GC-resistance in *E. coli* is confirmed as still ongoing globally [13]. As for Italy, the population-weighted mean resistance of 3GC resistant *E. coli* in 2019 was 30.5%, showing, if compared to the European mean, a high prevalence even if a stable trend since 2015 [14].

The causal relationship between antibiotic consumption and antibiotic resistance is described at both the individual and the community level [15,16]. In particular, the use of broad-spectrum antibiotics is fostering the community spread of resistant bacteria worldwide [17]. The association between antibiotic consumption and development of resistance is strongest when antibiotic use is recent, and studies show that resistance decreases over time. This is likely due to a functional trade-off between the maximal fitness in the presence or in the absence of an antibiotics within bacteria. What remains unclear is how much time it takes to restore antibiotic susceptibility and how this varies for different antibiotic classes and in different bacterial species [18]. The better understanding of antibiotic resistance decay over time in individual patients is relevant to inform public campaigns and to instruct physicians through targeted interventions aimed to promote a rational use of antibiotics in the community [18,19,20]. Therefore, the aim of this study was to estimate the impact of outpatient antibiotic use on the occurrence of infections caused by 3GCREC and the resistance decay over time. The primary objective was to determine the influence of previous antibiotic use on the development and decay of resistance to 3GCs of *E. coli* in patients treated by Primary Care Physicians. A secondary objective was to determine, in the same population of patients, the influence of previous 3GC use on the development and decay of resistance to 3GC of *E. coli*.

## 2. Results

A total of 1794 isolates were included in the analyses. Among them, urine isolates were the most frequent, followed by blood cultures as illustrated in Figure 1.

Within the study period, 241 cases and 1553 controls met the inclusion criteria, with a ratio of 6.44 controls per case. Patient characteristics are shown in Table 1.

Comparing cases and controls, the median age in case patients was 79 years while in controls it was 76 years (*p* = 0.003). The sample was predominantly composed of women (56% in cases and 66% in controls, *p* = 0.004). Moreover, case patients were more likely to be treated with many different drugs than controls (*p* < 0.0001), consistently with the finding that among these patients a higher burden of chronic diseases could be observed: case patients were more likely to be affected by diabetes (*p* < 0.0001), cancer (*p* = 0.012), COPD (*p* = 0.031) and end-stage kidney disease (*p* = 0.005) compared to control patients. In univariate analysis, 3GC-resistance was associated with longer hospital stays, hospitalization with surgical interventions and organ transplant, even if these associations were weak.

Concerning the influence of previous antibiotic use on current resistance of *E. coli* to 3GCs, an overall higher exposure to antibiotic drugs could be observed in patients who tested positive to 3GCREC compared to those infected with sensitive strains, showing a clear association between prior antibiotic use and the development of 3GC-resistance in *E. coli* at an individual level. Moreover, in univariate analysis the risk of colonization or infection due to 3GCREC was higher if at least one antibiotic prescription was issued to the patient in the year preceding hospitalization (OR 2.69, *p* < 0.0001), still elevated but decreasing in those patients in which the antibiotic therapy was undergone two years previous (OR 1.65, *p* < 0.0001) or three to five years (OR 1.59, *p* = 0.002) prior to hospitalization.

The final regression model included 10 variables which reached statistical significance (*p* < 0.05) in the univariate analysis (Table 2): age, gender, DDDs of drugs taken in the five years preceding hospitalization, number of antibiotics (J01) taken in the previous five years, at least one prescription of antibiotics (J01) taken in the previous 5, 4, 3 years, at least one prescription of antibiotics (J01) taken in the second previous year, at least one prescription of antibiotics (J01) taken in the previous year, days of hospital stay, diagnosis of diabetes.

The analysis showed that independent risk factors for being infected or colonized by 3GCREC are the following: male sex (OR 1.49, 95% CI 1.10–2.01, *p* = 0.009), older age (OR 1.11, 95% CI 1.02–1.21, *p* < 0.0001), the number of different antibiotics taken in the previous five years (OR 1.20, 95% CI 1.08–1.33, *p* < 0.013), at least one antibiotic (J01) prescription in the previous year (OR 1.92, 95% CI 1.36–2.71, *p* = 0.001), and the diagnosis of diabetes (OR 1.57, 95% CI 1.08–2.30, *p* = 0.019). Moreover, the analysis showed a significant, albeit weak association between longer hospital stays and 3GC-resistance of *E. coli* (OR 1.06, 95% CI 1.03–1.08, *p* < 0.0001).

Concerning the decay of the risk of resistance of *E. coli* isolates to 3GCs over time, the results of the univariate analysis are confirmed in the multivariate model, at least as a trend. As shown in Table 2 and in Figure 2, having received at least one antibiotic prescription three to five years before hospitalization was associated with a lower risk for patients of being colonized or infected with 3GC-resistant *E. coli* (OR 0.85, 95% CI 0.59–1.24, *p* = 0.399) than receiving an antibiotic prescription both in the second year before hospitalization (OR 0.97, 95% CI 0.68–1.38, *p* = 0.866) or in the year preceding hospitalization (OR 1.92, 95% CI 1.36–2.71, *p* < 0.001).

An analysis focused on the last year prior to hospitalization (Table 3) revealed a dose-response effect of antibiotic use on resistance: the use of 3GC increases the risk of being infected or colonized by 3GCREC more than two-fold if two or more prescriptions of 3GCs were issued in the considered period of time (OR 2.08, 95% CI 1.07–4.08, *p* = 0.030). This association turned to be protective in our sample when considering a lower exposure to 3GC, even if statistically not significant (OR 0.73, 95% CI 0.37–1.40, *p* = 0.345), showing that a cumulative exposure to 3GCs in the prior 12 months had a clear dose-response effect on 3GC-resistance in *E. coli*. Consistently, the same dose—response effect is observed when considering any antibiotic (J01) the exposure of interest (OR 2.03, 95% CI 1.45–2.85, *p* < 0.0001).

To ascertain the consistency of these findings, four sensitivity analyses were carried out: (1) to rule out an effect of in-hospital antibiotic administration or hospital acquired infections, all patients who were tested 48 h after their hospital admission were excluded (Appendix A); (2) to rule out the effect on susceptibility testing of very recent 3GC-use (we were interested in the effect of less recent antibiotic use), all patients who received at least one 3GC prescription 15 days prior to hospitalization were excluded (Appendix A); (3) to rule out a potential effect of co-resistance, controls were defined as those subjects who have had a bacterial culture with *E. coli* sensitive to any antibiotic and not only to 3GCs (Appendix A). All four sensitivity analyses showed results consistent with those of the main analyses.

## 3. Discussion

### 3.1. Summary of Main Findings

The present case-control study is the first one to use routinely collected healthcare data in a multiple-database approach that characterize factors associated with community-acquired antibiotic resistant infections in Italy. Moreover, to the best of our knowledge, it is the first study that provides evidence about resistance decay in individuals after outpatient antibiotic use using long term data. We found that, over a 5-year period, the risk of developing a community acquired infection due to 3GCREC increases significantly in patients who were exposed to antibiotics previously, with the highest risk observed for antibiotics taken in the last 12 months and for greater cumulative exposures to any antibiotic as well as to 3GCs. Apart from previous antibiotic use, we also found male sex, older age, and the presence of diabetes to be significantly associated with 3GC-resistance of *E. coli* after adjustment for other factors.

### 3.2. Comparison to Existing Literature

Our results are consistent with a recent study on 146,452 *E. coli* isolates from 143 tertiary care hospitals in China. Authors found that 3GC-resistance of *E. coli* correlated with the prior consumption of all antibiotics, as well as, specifically, of β-lactams, including cephalosporins and 3GCs [21]. However, the study was not designed to adjust these associations with respect to other factors, nor did it take into account the duration of the association with respect to the interval between antibiotic consumption and the diagnosis of resistance. A systematic review of five randomized controlled trials and 19 observational studies from 2010 [16] found that the association between antibiotic consumption and resistance was strongest at 0–1 months from exposure and could last for up to 12 months. None of the included studies analyzed specifically the association between antibiotic use and 3GC-resistance of *E. coli*, the majority focusing on single antibiotics rather than all antibiotics [22,23] or used interviews rather than actual prescriptions to estimate the exposure variables [24]. A more recent systematic review of five randomized controlled trials and 20 prospective observational studies from 2018 [18] found that AMR was highest soon after antibiotic use and showed a decrease of resistance after 1–3 months, a faster decrease for at least one of the bacteria (penicillin-resistant *Streptococcus pneumoniae*) than previously reported.

Our results are consistent with the findings from Costelloe et al. [16] and Bakhit et al. [18] but, in addition, we were able to measure a long term trend in the decay of resistance by examining the 5 year period prior to the diagnosis of the resistant infection. Moreover, we found that cumulative exposure to 3GCs is an independent risk factor for being diagnosed with community acquired 3GCREC.

According to a recent study from Taiwan focusing on factors associated with community acquired 3GCREC infections, 3GC-resistance in *E. coli* is an independent risk factor for longer hospital stays [25]. A similar finding was reported also in previous studies [26,27]. In our study we could detect a significant, albeit weak, association between longer hospital stays and 3GC-resistance in *E. coli* (OR 1.06, 95% CI 1.03–1.08, *p* < 0.0001). This association could be interpreted as follows: longer hospital stays could be an independent risk factor for 3GCREC colonization/infection, although we cannot exclude that *E. coli* resistant to 3GCs is, vice versa, an independent risk factor for longer hospital stays. This because resistance to 3GC was diagnosed during the hospital stay.

Resistance to 3GCs in Enterobacteriaceae is frequently caused by Extended Spectrum Beta-lactamase (ESBL)-producing bacteria [28,29]. ESBL-producing isolates often show resistance to other β-lactams, and can be associated also with aminoglycoside and fluoroquinolone resistance [30]. A recent systematic review of 27 observational studies on risk factors of fluoroquinolone-resistance in *E. coli* found that previous antibiotic use was a strong independent risk factor for resistance (OR 2.74, 95% CI 1.92 to 3.92). Fluoroquinolone-resistance in *E. coli* was reported as independently associated with diabetes mellitus (OR 1.62, 95% CI 1.43 to 1.83) and male sex (OR 1.41, 95% CI 1.21 to 1.64). Moreover, studies on community acquired ESBL-producing *E. coli* infections have identified diabetes mellitus (but not male sex) as an independent risk factor for these kind of infections [31,32]. Our study suggests that both diabetes mellitus and male sex are risk factors for community acquired *E. coli* resistance to 3GCs, although more primary research is needed to better define risk factors of 3GC-resistance in *E. coli*.

### 3.3. Strengths and Limitations

A significant strength of this study is the data source: (1) hospital discharge records include data from every single hospital discharge carried out in the given period of time; (2) the database of drug prescription records contains any antibiotic prescription covered by the NHS, filled by every single physician in the examined period of time; (3) databases of the regional reference laboratories contain all susceptibility tests performed in the province where this study was carried out. This allowed us a data linkage among comprehensive datasets covering a catchment area 532,644 inhabitants [33]. Furthermore, the use of data from multiple health information systems made it possible to collect information on exposure from months or even years earlier without the risk of recall bias, as well as data on hospitalizations and comorbidities.

Some limitations have to be mentioned. We were able to establish the exact date of diagnosis of bacterial resistance, but we know nothing about the actual date of the onset of the resistance. Thus, the association between antibiotic consumption and resistance could be interpreted in two ways: (a) the high level of antibiotic use induces resistance in the bacteria; (b) the high level of antibiotic use is a consequence of non-response to antibiotic therapy in individuals already colonized by resistant bacteria.

Another limitation of the study is that the DDD gives a rough estimate of drug consumption and reflects only approximately the dose and the length of treatment. Moreover, we assume that the antibiotics prescribed were actually taken. If this were not the case, there would be an overestimation of exposure. Finally, we have no information on privately purchased antibiotics, which could lead to an underestimation of exposure. However, in Italy the rate of privately purchased antibiotics that cannot be tracked is 17.4% of total outpatient antibiotic consumption, with 3GC representing less than 0.1% [9]. Thus, if anything, it is probable that underestimation occurred for the exposure to other more used antibiotic rather than to 3GCs.

## 4. Materials and Methods

### 4.1. Study Design, Setting and Data Sources

The present case-control study was conducted in the Autonomous Province of Bolzano/Bozen, located in northern Italy, which at the date of 1 January 2020 accounts for 532,644 inhabitants [34], under a scientific agreement with the Italian National Institute of Health (ISS) with the aim of conducting pharmacoepidemiology studies on large databases by linking different sources of routinely collected health data. Data were obtained from the following information systems:The database of hospital laboratories of the Autonomous Province of Bolzano/Bozen that was used to define cases and controls.The database of outpatient pharmaceutical prescriptions of the Bolzano/Bozen Local Health Trust, that was used to define the exposure.The hospital discharge record database of the Autonomous Province of Bolzano/Bozen, that was used to identify potential risk factors.

The database of hospital laboratories was queried to extract all patients who were hospitalized in 2016 and for whom a bacterial culture test was carried out. From those, only patients with *E. coli* isolates were included in the study and patients were classified as cases, if they carried 3GC-resistant isolates, or controls, if they carried 3GC-sensitive isolates. For each case, we matched all available controls on the date of the first isolate (±30 days). We defined the sampling date of the first bacterial isolate as the “index date” and we cleared cases and controls for which the index date was not available. If, during 2016, a patient had more than one laboratory report attesting either bacterial resistance or negative isolates, only the first one was used. Figure 3 represents a flow-diagram of included cases and controls.

Analyses were conducted on the following biological materials: blood, urine, respiratory tract secretions, soft tissue specimens and various others (including vulvar, vaginal and perianal specimens, ascites and other abdominal fluid, pleural liquid and post-surgery drainage fluid). All specimens were processed in the laboratories of the hospitals included in the study. Bacterial species were identified using the VITEK II system (bioMérieux, Hazelwood, MO, USA) or the Matrix-assisted-laser-desorption-ionization time-of-flight mass spectrometry (Maldi-TOF). Antimicrobial susceptibility testing was performed using the VITEK II system. The interpretation of the antibiograms was based on the European Committee on Antimicrobial Susceptibility Testing (EUCAST) interpretation criteria (http://www.eucast.org/, accessed on 27 February 2021). Results of the performed antibiograms were classified as follows: Resistant (R), sensible (S) or intermediate (I). Only patients carrying isolates with a R or S test results were included in the analyses.

The database of outpatient pharmaceutical prescriptions of the Bolzano/Bozen Health Service, which contains all prescriptions covered by the National Health System (NHS) and is regularly updated, was queried to extract all pharmaceutical prescriptions issued from 1 January 2011 to 31 December 2016 by primary care physicians of the Province of Bolzano/Bozen (i.e., general practitioners and out of hours primary care physicians). Prescriptions issued by hospital-based physicians were excluded in order to meet the study objectives, namely, to point out the impact on resistance of outpatient antibiotic use. Antibiotics used in the 5 years preceding the index date were categorized according to the Anatomical Therapeutic Chemical (ATC) (https://www.whocc.no/, accessed on 27 February 2021) classification system: J01 for general antibiotics and J01DD for 3GC (Appendix A) [35].

We also used cumulative define daily doses (DDDs) of different drug classes taken in the previous five years, as a comorbidity measure. The correspondence between medications classified according to ATCs and International Classification of Diseases, Ninth Revision, Clinical Modification (ICD-9-CM) is shown in the Appendix A (Appendix A) [36].

The hospital discharge records database of the Autonomous Province of Bolzano/Bozen contains the discharge records of all the hospitals of the Province. The information collected includes patient demographics, hospitalization characteristics (e.g., discharge date, hospitalization regimen, discharge modalities) and clinical characteristics (e.g., primary diagnosis, concomitant diagnosis, diagnostic or therapeutic procedures). It was used to assess potential risk factors for colonization or infection with 3GC-resistant *E. coli* that could act as confounding factors, namely: age, gender, the hospital ward in which the patient was admitted, total days of hospitalizations, hospitalizations with surgery, hospitalizations with device implantation, hospitalizations with organ transplant, diagnosis of chronic diseases (cancer, diabetes, chronic obstructive pulmonary disease [COPD], AIDS, immunosuppression), hemodialysis and previous use of cortisone drugs.

Potential risk factors for 3GC-resistance in *E. coli* and their data sources are listed Table 4.

### 4.2. Definition of Exposure

Subjects were considered exposed if they received at least one prescription of antibiotics (ATC J01) issued by a primary care physician in the 5 years preceding the hospitalization in which a bacterial culture test was carried out. In order to meet the secondary objective of the study, exposed subjects were considered only those who were prescribed with a 3GC (see Appendix A) in the previous 12 months.

### 4.3. Statistical Analysis

Descriptive analyses were conducted to compare the characteristics of enrolled patients, namely of cases and controls. Categorical variables were presented as percentage, while continuous variables were reported as mean (±standard deviation) or, where appropriate, as median (interquartile range). Risk factors for 3GC-resistance were analyzed using conditional logistic regression. Matched Odds ratios (ORs) and 95% confidence intervals (CIs) were calculated to evaluate the strength of any association that emerged. Factors with a *p*-value < 0.05 in the univariate analysis were considered eligible for the multivariate analysis and was included using a backward stepwise selection method.

The following sensitivity analyses were carried out.

Firstly, due to the fact that the aim of the present study was to ascertain the effect on bacterial resistance of community antibiotic use over time, we excluded from the analyses all patients with an index date 48 h away from the hospitalization date. This in order to exclude all potential hospital acquired infections from the outcome measure.

Secondly, in order to eliminate the potential effect of very recent antibiotic use on bacterial resistance, the second sensitivity analysis was performed excluding all patients who received at least one prescription of antibiotics in the 15 days preceding hospitalization.

Lastly, a sensitivity analysis used as controls only subjects with an *E. coli* isolate fully sensitive to all antibiotics. This in order to in order to exclude a potential confounding effect of co-resistance.

All the analyses were performed using STATA software package version 13.0 [37] and R 3.6 [38].

## 5. Conclusions

Over a 5-year period, we found that the risk of developing a community acquired infection due to 3GCREC increases significantly in patients who were exposed to antibiotics previously, with the highest risk observed for greater cumulative exposures to any antibiotic as well as to 3GC. We found that the effect of antibiotic exposure on 3GC-resistance of *E. coli* was highest in the first 12 months after antibiotics were taken and then decreased progressively.

These findings can be useful to inform public campaigns and to instruct physicians through targeted interventions aimed to promote a rational use of antibiotics in the community, although more studies are needed on different bacterial species and antibiotic classes from more long-term data.

## Figures and Tables

**Figure 1 antibiotics-10-00451-f001:**
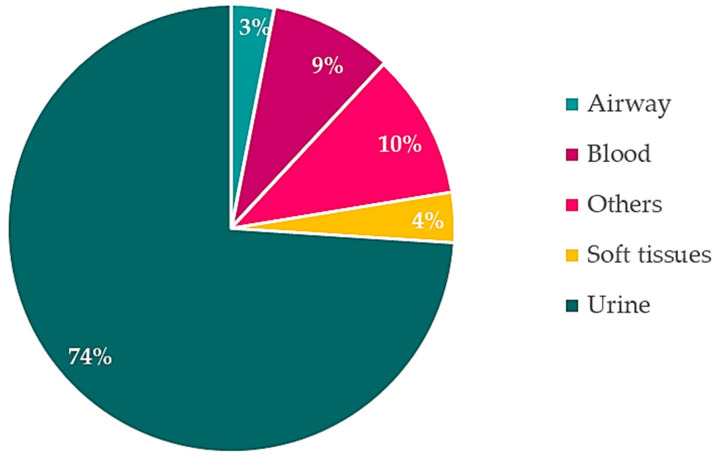
Distribution of included isolates. Others = vulvar, vaginal and perianal specimens, ascites and other abdominal fluid, pleural liquid, post-surgery drainage fluid.

**Figure 2 antibiotics-10-00451-f002:**
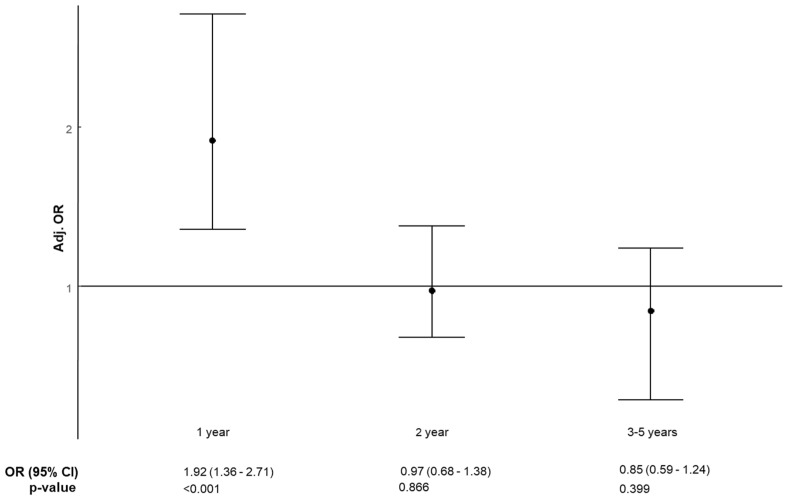
Forest plot showing the influence of previous antibiotic use in the past 5 years on the decay of resistance to third generation cephalosporins (3GCs) of *E. coli* over time (multivariate analysis).

**Figure 3 antibiotics-10-00451-f003:**
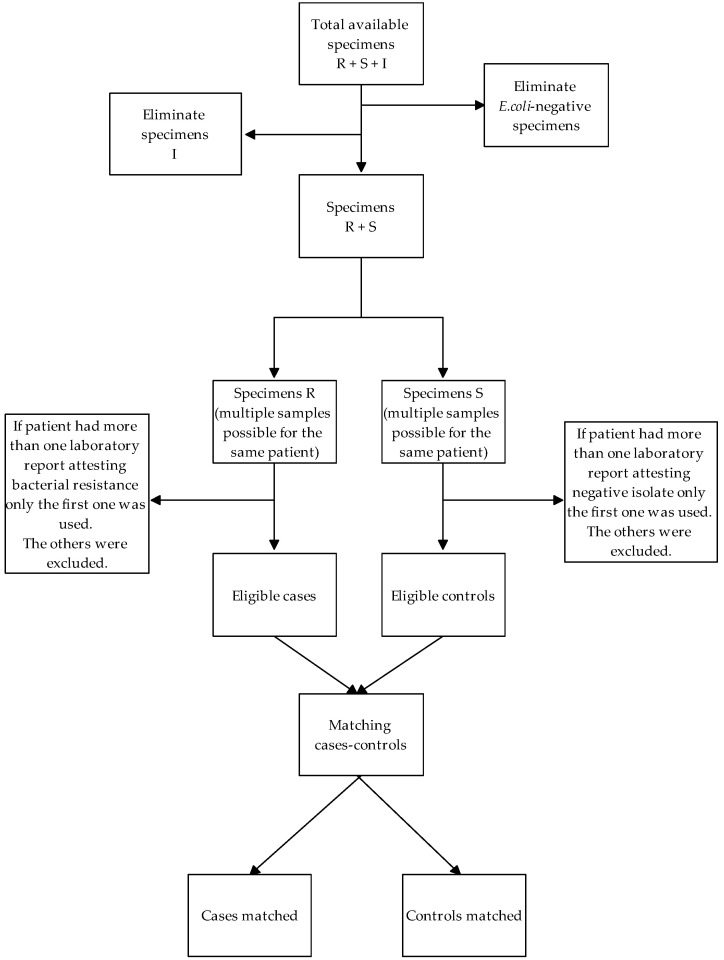
Flow diagram of included cases and controls. Abbreviations: R = 3GC-resistant specimens; S = 3GC-sensible specimens; I = 3GC-intermediate specimens.

**Table 1 antibiotics-10-00451-t001:** Characteristics of 241 case patients infected/colonized with 3GC-resistant *E. coli* and 1553 matched control patients infected/colonized with 3GC-susceptible *E. coli*.

Variable	Cases ^a^	Controls ^a^	Crude OR (95% CI)	*p*
Age, Median (IQ)	79 (68–85)	76 (61–84)	1.11 ^b^ (1.04–1.20)	0.003
Gender				
Male (%)	106 (43.98)	530 (34.13)	1.49 (1.14–1.98)	0.004
Hospitalization ward				
Other (%)	4 (1.66)	12 (0.77)	-	
Surgery (%)	82 (34.02)	597 (38.44)	0.44 (0.14–1.41)	0.167
Intensive care unit (%)	12 (4.98)	78 (5.02)	0.47 (0.13–1.71)	0.252
Medicine unit (%)	143 (59.34)	866 (55.76)	0.54 (0.17–1.71)	0.296
Drug’s DDD taken in previous 5 years, Median (IQ)	4334.15 (1092.92–7802.62)	3117.01 (414.83–6392.25)	1.04 ^c^ (1.01–1.06)	0.016
Number of active ingredients taken in previous 5 years, Median (IQ)	16 (9–24)	12 (5–19)	1.04 (1.03–1.05)	<0.0001
At least one cortisone drug’s DDD taken in previous 5 years (%)	29 (12.03)	139 (8.95)	1.38 (0.90–2.13)	0.144
Number of antibiotics taken in previous 5 years, Median (IQ)	3 (1–4)	2 (0–3)	1.27 (1.19–1.35)	<0.0001
At least one J01 prescription taken in previous period (%)				
I year	161 (66.80)	681 (43.85)	2.69 (2.00–3.61)	<0.0001
II year	127 (52.70)	626 (40.31)	1.65 (1.26–2.17)	<0.0001
III–V years	161 (66.80)	879 (56.60)	1.59 (1.19–2.12)	0.002
At least one 3GC prescription in previous year (%)				
0	212 (87.97)	1437 (92.53)	-	-
1	12 (4.98)	78 (5.02)	1.04 (0.56–1.95)	0.898
2+	17 (7.05)	38 (2.45)	3.23 (1.78–5.87)	<0.0001
Hospitalization days, Median (IQ)	47 (8–97)	12 (0–46)	1.05 ^d^ (1.04–1.07)	<0.0001
Hospitalization with surgery (%)	177 (48.55)	642 (41.34)	1.32 (1.01–1.73)	0.044
Hospitalization with device implantation (%)	29 (12.03)	127 (8.18)	1.52 (0.98–2.34)	0.059
Hospitalization with organ transplant (%)	6 (2.49)	26 (1.67)	1.59 (0.65–3.88)	0.305
Diagnosis of chronic diseases				
Cancer (%)	59 (24.48)	270 (17.51)	1.52 (1.10–2.11)	0.012
Diabetes (%)	61 (25.31)	251 (16.16)	1.79 (1.29–2.47)	<0.0001
AIDS (%)	0 (0.00)	2 (0.13)	NA	NA
COPD (%)	86 (35.68)	461 (29.68)	1.39 (1.03–1.87)	0.031
Immunosuppression (%)	0 (0.00)	2 (0.13)	NA	NA
Haemodialysis (%)	7 (2.90)	29 (1.87)	1.22 (1.06–1.40)	0.005

^a^ Number (%) of patients or median (IQ); ^b^ OR calculated for 10-year increments; ^c^ OR calculated for 1000-DDD increments; ^d^ OR calculated for 10-day increments. Abbreviations: IQ = interquartile range; DDD = defined daily dose; AIDS = acute immune deficiency syndrome; COPD = chronic obstructive pulmonary disorder; NA = not Applicable.

**Table 2 antibiotics-10-00451-t002:** Summarized results of the main multivariate analysis (backward step-wise approach).

Variables	Cases ^a^	Controls ^a^	Adj OR (95% CI)	*p*
Age, Median (IQ)	79 (68–85)	76 (61–84)	1.11 ^b^ (1.02–1.21)	<0.0001
Gender, Male (%)	106 (43.98)	530 (34.13)	1.49 (1.10–2.01)	0.009
Drug’s DDD taken in previous 5 years, Median (IQ)	4334.15 (1092.92–7802.62)	3117.01 (414.83–6392.25)	0.95 ^c^ (0.91–0.99)	0.013
Number of antibiotics taken in previous 5 years, Median (IQ)	3 (1–4)	2 (0–3)	1.20 (1.08–1.33)	0.001
At least one J01 prescription taken in previous period (%)				
I year	161 (66.80)	681 (43.85)	1.92 (1.36–2.71)	<0.0001
II year	127 (52.70)	626 (40.31)	0.97 (0.68–1.38)	0.866
III—V years	161 (66.80)	879 (56.60)	0.85 (0.59–1.24)	0.399
Hospitalization days, Median (IQ)	47 (8–97)	12 (0–46)	1.06 ^d^ (1.03–1.08)	<0.0001
Hospitalizations with surgery (%)	177 (48.55)	642 (41.34)	0.89 (0.82–0.98)	0.012
Diabetes (%)	61 (25.31)	251 (16.16)	1.57 (1.08–2.30)	0.019

^a^ Number (%) of patients or median (IQ); ^b^ OR calculated for 10-year increments; ^c^ OR calculated for 1000-DDD increments; ^d^ OR calculated for 10-day increments. Abbreviations: IQ = interquartile range; DDD = defined daily dose.

**Table 3 antibiotics-10-00451-t003:** Summarized results of multivariate analysis (backward step-wise approach) focused on 3GC use in the 12 months preceding hospitalization.

Variables	Cases ^a^	Controls ^a^	Adj OR (95% CI)	*p*
Age, Median (IQ)	79 (68–85)	76 (61–84)	1.11 ^b^ (1.02–1.21)	0.012
Gender, Male (%)	106 (43.98)	530 (34.13)	1.49 (1.10–2.02)	0.010
Drug’s DDD taken in previous 5 years, Median (IQ)	4334.15 (1092.92–7802.62)	3117.01 (414.83–6392.25)	0.95 ^c^ (0.91–0.99)	0.009
Number of antibiotics taken in previous 5 years, Median (IQ)	3 (1–4)	2 (0–3)	1.15 (1.05–1.25)	0.002
At least one other J01 prescription in previous year (%)	156 (64.73)	640 (41.21)	2.03 (1.45–2.85)	<0.0001
3GC prescriptions in the previous year (%)				
0	212 (87.97)	1437 (92.53)	-	-
1	12 (4.98)	78 (5.02)	0.73 (0.37–1.40)	0.345
2+	17 (7.05)	38 (2.45)	2.08 (1.07–4.08)	0.030
Hospitalization days, Median (IQ)	47 (8–97)	12 (0–46)	1.06 (1.03–1.08)	<0.0001
Hospitalization with surgery (%)	177 (48.55)	642 (41.34)	0.89 ^d^ (0.82–0.99)	0.010
Diabetes (%)	61 (25.31)	251 (16.16)	1.6 (1.09–2.34)	0.016

^a^ Number (%) of patients or median (IQ); ^b^ OR calculated for 10-year increments; ^c^ OR calculated for 1000-DDD increments; ^d^ OR calculated for 10-day increments. Abbreviations: IQ = interquartile range; DDD = defined daily dose, 3GC = third generation cephalosporin.

**Table 4 antibiotics-10-00451-t004:** Potential risk factors for 3GC-resistance in *E. coli* and their data source.

Potential Confounding Factor	Data Source
Age	
Gender	hospital discharge records database
Hospitalization ward	hospital discharge records database
Surgery	hospital discharge records database
Intensive care unit	
Internal Medicine	
Other	
Drug’s DDD taken in previous 5 years	
Number of active ingredients taken in previous 5 years	database of drug prescription records
One or more cortisone drug DDDs taken in previous 5 years	database of drug prescription records
Number of antibiotics taken in previous 5 years	database of drug prescription records
One or more J01 prescription taken in previous 5,4,3 years	
One or more J01 prescription taken in previous 2 years	database of drug prescription records
Hospitalization days	database of drug prescription records
Hospitalizations with surgery	
Hospitalizations with device implantation	database of drug prescription records
Hospitalizations with organ transplant	hospital discharge records database
Diagnosis of chronic diseases	hospital discharge records database
Cancer	hospital discharge records database
Diabetes Mellitus	hospital discharge records database
AIDS	hospital discharge records database
COPD	
Immunosuppression	
Hemodialysis	

Abbreviations: DDD = defined daily dose; AIDS = acute immune deficiency syndrome; COPD = chronic obstructive pulmonary disorder.

## Data Availability

The raw data are available upon reasonable request from the corresponding author.

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
