# Peer review of "Impact of Prior Antibiotic Use in Primary Care on Escherichia coli Resistance to Third Generation Cephalosporins: A Case-Control Study"

_antibiotics, 2021, doi:10.3390/antibiotics10040451_

Round 1

Reviewer 1 Report

A well conducted retrospective study resulting in significant final conclusion for practitioners. No methodical comments and just a minor text editing is needed.

Author Response

Thank you for your review. We really appreciated that you found our study solid and useful. We went through the manuscript carefully and revised the text errors accordingly. The changes are listed as tracked changes in the revised version.

Regards

Peter Kurotschka

Reviewer 2 Report

A very nice paper describing the impact of prior antibiotic use and the development of third-generation cephalosporins. The introduction is well written and provides the reader with the context and detail for the results which are presented well and give the reader the data required to understand the discussion. The discussion provides an interesting summary and comparison to the literature data which seem to support the results here. Finally, the methods provided give a clear indication of how the data/results were generated. 

Barring some minor English spelling changes and amendments I would recommend publication of this article.

Author Response

Thank you for your review. We really appreciated that you found our study well written. We went through the manuscript carefully and revised the typos and a few other grammatical mistakes. The changes are listed as tracked changes in the revised version.

Best regards

Reviewer 3 Report

I find this paper very interesting. Nowadays, the prominent role of microorganisms and their role in our bodies are increasingly day by day in primary care and scientific communities. In the introduction, the discovery of antibiotics, as authors well commented, supposed an evolution for the life of humans, but in the same way, years later, their same excessive use causes us damages that we still do not know and want to solve. In this part, you could mention more number of references of the benefits of antibiotics, and at another point/paragraph I would contine with problems emerging from them. You can check:

  • Biswas R, Halder U, Kabiraj A, Mondal A, Bandopadhyay R. Overview on the role of heavy metals tolerance on developing antibiotic resistance in both Gram-negative and Gram-positive bacteria. Arch Microbiol. 2021 Apr 2. doi: 10.1007/s00203-021-02275-w. Epub ahead of print. PMID: 33811263.
  • Johnsen PJ, Gama JA, Harms K. Bacterial evolution on demand. Elife. 2021 Apr 6;10:e68070. doi: 10.7554/eLife.68070. PMID: 33820602.
  • MacKinnon MC, Sargeant JM, Pearl DL, Reid-Smith RJ, Carson CA, Parmley EJ, McEwen SA. Evaluation of the health and healthcare system burden due to antimicrobial-resistant Escherichia coli infections in humans: a systematic review and meta-analysis. Antimicrob Resist Infect Control. 2020 Dec 10;9(1):200. doi: 10.1186/s13756-020-00863-x. PMID: 33303015; PMCID: PMC7726913.

The paper was focused on the thirg generation cephalosporin resistant E.coli (3GCREC), they mentioned that these bacteria are among the top ranked of cases that are attributed to death. You must indicate more references regarding this aim or complement the statement with the species of batería that are within the most relevant list, since other types of microorganisms families are mentioned in the discussion. You can see these references:

  • Hiltunen T, Virta M, Laine AL. Antibiotic resistance in the wild: an eco-evolutionary perspective. Philos Trans R Soc Lond B Biol Sci. 2017 Jan 19;372(1712):20160039. doi: 10.1098/rstb.2016.0039. PMID: 27920384; PMCID: PMC5182435.
  • Ploy MC, Lambert T, Gassama A, Denis F. Place des intégrons dans la dissémination de la résistance aux antibiotiques [The role of integrons in dissemination of antibiotic resistance]. Ann Biol Clin (Paris). 2000 Jul-Aug;58(4):439-44. French. PMID: 10932044.
  • Escudero JA, Loot C, Nivina A, Mazel D. The Integron: Adaptation On Demand. Microbiol Spectr. 2015 Apr;3(2):MDNA3-0019-2014. doi: 10.1128/microbiolspec.MDNA3-0019-2014. PMID: 26104695.

In results, I am really impressive with the data, but also I want suggest that you can use some stadistical tool whose can measure if the information of patients that you have collected according to the hospital reports correspond to the same number of patients that they could refer you if they have received these treatments or completed the dosis. The authors commented it in limitations (line 242) but if they could be see a bit part of the data that they used, maybe they can establish or observe guidance on level of control in hospital reports as well as the level of relevance of using this type of database for these studies, especially when information cannot be obtained directly from patients. Also, if it were possible it would be a very important part to be able to analyze whether the number of patients with 3GCREC also show through the 16S ribosomal RNA or DNA analysis that these bacteria have been identified and classified correctly in the samples that they can isolated. You can see:

  • Gamage HKAH, Venturini C, Tetu SG, Kabir M, Nayyar V, Ginn AN, Roychoudhry B, Thomas L, Brown M, Holmes A, Partridge SR, Seppelt I, Paulsen IT, Iredell JR. Third generation cephalosporins and piperacillin/tazobactam have distinct impacts on the microbiota of critically ill patients. Sci Rep. 2021 Mar 31;11(1):7252. doi: 10.1038/s41598-021-85946-4. PMID: 33790304; PMCID: PMC8012612.
  • Meyer E, Lapatschek M, Bechtold A, Schwarzkopf G, Gastmeier P, Schwab F. Impact of restriction of third generation cephalosporins on the burden of third generation cephalosporin resistant K. pneumoniae and E. coli in an ICU. Intensive Care Med. 2009 May;35(5):862-70. doi: 10.1007/s00134-008-1355-6. Epub 2008 Nov 26. PMID: 19034426.

In table 1 (line 130), you showed data of patients, I am checking that the average is elderly people, you must metion more references about microorganisms in body changes while we are growing or broadly in different age-groups or by changes in diet or diet and activity profile. Also, in the pharmaceuticals treatments you did not description any antihypertensive drug, probably patients consume them, but if you considered that this information is not relevant for this table you must explained it or mentioned at least.

In table 2, (line 156-158), you showed all p-values in table but I think that it is better if you indicates in the paragraph too, looks like more clear if you can check tables fastly or something can be happen in this way, especially when some of your ORs oscillate in ranges lower than 1 and higher.

In table 3, line 163, I am curious with 3GC prescription in previous year % in 1 year, the OR oscillates between protection and risk values ​​and also the p-value is not significant, but at 2 years or more is clearly numbers of risk values, maybe you can explain something more about it.

Author Response

Thank you for your interesting and inspirational suggestions. Please find our replies below your main points.

I find this paper very interesting. Nowadays, the prominent role of microorganisms and their role in our bodies are increasingly day by day in primary care and scientific communities. In the introduction, the discovery of antibiotics, as authors well commented, supposed an evolution for the life of humans, but in the same way, years later, their same excessive use causes us damages that we still do not know and want to solve. In this part, you could mention more number of references of the benefits of antibiotics, and at another point/paragraph I would contine with problems emerging from them. You can check:

  • Biswas R, Halder U, Kabiraj A, Mondal A, Bandopadhyay R. Overview on the role of heavy metals tolerance on developing antibiotic resistance in both Gram-negative and Gram-positive bacteria. Arch Microbiol. 2021 Apr 2. doi: 10.1007/s00203-021-02275-w. Epub ahead of print. PMID: 33811263.
  • Johnsen PJ, Gama JA, Harms K. Bacterial evolution on demand. Elife. 2021 Apr 6;10:e68070. doi: 10.7554/eLife.68070. PMID: 33820602.
  • MacKinnon MC, Sargeant JM, Pearl DL, Reid-Smith RJ, Carson CA, Parmley EJ, McEwen SA. Evaluation of the health and healthcare system burden due to antimicrobial-resistant Escherichia coli infections in humans: a systematic review and meta-analysis. Antimicrob Resist Infect Control. 2020 Dec 10;9(1):200. doi: 10.1186/s13756-020-00863-x. PMID: 33303015; PMCID: PMC7726913.

The paper was focused on the third generation cephalosporin resistant E.coli (3GCREC), they mentioned that these bacteria are among the top ranked of cases that are attributed to death. You must indicate more references regarding this aim or complement the statement with the species of batería that are within the most relevant list, since other types of microorganisms families are mentioned in the discussion. You can see these references:

  • Hiltunen T, Virta M, Laine AL. Antibiotic resistance in the wild: an eco-evolutionary perspective. Philos Trans R Soc Lond B Biol Sci. 2017 Jan 19;372(1712):20160039. doi: 10.1098/rstb.2016.0039. PMID: 27920384; PMCID: PMC5182435.
  • Ploy MC, Lambert T, Gassama A, Denis F. Place des intégrons dans la dissémination de la résistance aux antibiotiques [The role of integrons in dissemination of antibiotic resistance]. Ann Biol Clin (Paris). 2000 Jul-Aug;58(4):439-44. French. PMID: 10932044.
  • Escudero JA, Loot C, Nivina A, Mazel D. The Integron: Adaptation On Demand. Microbiol Spectr. 2015 Apr;3(2):MDNA3-0019-2014. doi: 10.1128/microbiolspec.MDNA3-0019-2014. PMID: 26104695.

We appreciate your stimulating comments about the possible consequences of antibiotic use and misuse for both human health and the environment which go beyond the more obvious effect in promoting AMR. We agree that this is a very interesting topic which deserves attention. Following your suggestion, we have slightly expanded the text and added the most appropriate of your suggested references (please see Introduction, lines 48-52). Moreover, we have added the reference MacKinnon et al, 2020 (now ref. 10), regarding the importance of antimicrobial-resistant E. coli infections for the health and healthcare system and complemented our statement including some other bacterial species that are listed as relevant bacteria for AMR (please see Introduction, lines  67-71). Our study, however, is more narrowly focused and deals with the risk of selecting resistance upon previous antibiotic clinical treatments and that is why we emphasized this aspect in the introduction. 

We find that, although also very interesting, the topic of integrons (and other mobile genetic elements) involved in driving resistance through HGT is outside the scope of this manuscript and should be treated in a dedicated analysis.

In results, I am really impressive with the data, but also I want suggest that you can use some stadistical tool whose can measure if the information of patients that you have collected according to the hospital reports correspond to the same number of patients that they could refer you if they have received these treatments or completed the dosis. The authors commented it in limitations (line 242) but if they could be see a bit part of the data that they used, maybe they can establish or observe guidance on level of control in hospital reports as well as the level of relevance of using this type of database for these studies, especially when information cannot be obtained directly from patients.

Thank you for the opportunity to comment on this point. The present study is a population-based matched case control study. Data were extracted from routine healthcare databases and do not contain more details on, for example, treatment duration or doses. This is the reason why, in pharmacoepidemiological studies, the DDDs are used as a globally established proxy to evaluate duration and doses of treatment that makes it feasible to perform such studies without collecting data asking specifically every involved patient individually. This is a strength of these kind of studies: recall bias is avoided and large sample sizes can be reached with a fraction of the resources necessary if patients were asked or clinical records were analyzed individually. As stated in the discussion, a limitation is that a drug prescription does not mean that the patient actually received the drug, for example because of poor compliance or other reasons. Despite this, misclassification resulting from this is probably low (consider, for example, that the ECDC rely on the same data source to estimate antibiotic consumption annually https://www.ecdc.europa.eu/en/antimicrobial-consumption/surveillance-and-disease-data/database). Moreover, other type of biases are avoided, such as recall bias, which could lead to misclassification especially in the context of studies that explore long term data, such as our study. An example of validation of administrative databases for research purposes in Italy is the VALORE project (https://doi.org/10.1371/journal.pone.0095419).

Also, if it were possible it would be a very important part to be able to analyze whether the number of patients with 3GCREC also show through the 16S ribosomal RNA or DNA analysis that these bacteria have been identified and classified correctly in the samples that they can isolated.

Analysis of 16S rRNA or DNA were not obtained in this study. However, if you mean that this analysis would be an important part of the work to rule out that the bacterial clinical isolates were misidentified and misclassified, our answer is that all the identifications and classifications were conducted at the Health Service of Bolzano/Bozen in regional reference laboratories, according to internationally accepted clinical laboratory standards (please see Material and Methods, lines 320-331) and, for this reason, we do not expect that this would be the case. If instead you mean that 16S rRNA or DNA analysis could have improved our work in the understanding the change of the microbiota, including 3GCREC, we agree that this would have been a very interesting aspect which, however, is beyond the scope of the present study.

In table 1 (line 130), you showed data of patients, I am checking that the average is elderly people, you must metion more references about microorganisms in body changes while we are growing or broadly in different age-groups or by changes in diet or diet and activity profile.

As mentioned above, we agree that discussing about a change in microbiota in different age-groups and also due to diet or other factors would have been desirable but, again, we think that this interesting aspect is beyond the scope of the present study and just to mention it without supportive data would be rather speculative at this point and, somewhat, distracting with respect to the focus of the work and the overall conclusions.

Also, in the pharmaceuticals treatments you did not description any antihypertensive drug, probably patients consume them, but if you considered that this information is not relevant for this table you must explained it or mentioned at least.

Thank you for the opportunity to comment also on this point. We used ATC codes (please see supplementary file) to cross-check comorbidities, which are more likely associated with a risk of infection. This is explained in the methods section (line 321 and followings). The information on antihypertensive drugs has no direct relationship with the research question, therefore we did not mention them in the results section. We believe that to mention it would not add anything to the work.

In table 2, (line 156-158), you showed all p-values in table but I think that it is better if you indicates in the paragraph too, looks like more clear if you can check tables fastly or something can be happen in this way, especially when some of your ORs oscillate in ranges lower than 1 and higher.

Thank you for this comment. We agreed, and have revised this throughout the text to include now the p-values, along with the confidence intervals.

In table 3, line 163, I am curious with 3GC prescription in previous year % in 1 year, the OR oscillates between protection and risk values ​​and also the p-value is not significant, but at 2 years or more is clearly numbers of risk values, maybe you can explain something more about it.

Table 3 shows the multivariate analysis focused on the last year prior to hospitalization. The variables of interest you were referring to are: 1) the use (i.e., prescription) of 1 course of 3GC, 2) the use of 2 or more courses of 3GC; 3) non-use (reference). The analysis showed the so-called dose-response effect: a lower cumulative exposure leads to smaller effects on the outcome than higher cumulative exposures. In our case, we have no statistically significant association between the lower “dose” (i.e., 1 course of treatment) of 3GC in the year prior to hospitalization and 3GC-resistance; the association becomes significant if 2 or more prescriptions were issued.

Following your suggestion, to make this clearer, we have modified the text to reflect that we refer to number of prescriptions "in the previous year" only (Table 3, left column). 

Best regards